# Digital economy, industrial structure upgrading and green total factor productivity —Evidence in textile and apparel industry from China

**Xiangmei Zhu, Bin Zhang** [ORCID] **\*, Hui Yuan**

School of Economics and Management, North University of China, Taiyuan, Shanxi, China

\* 1158597163@qq.com

## Abstract

According to the standard of GB/T4754-2017 Classification of National Economic Industry and the characteristics of the textile and apparel industry, the textile and apparel industry is divided into three categories: textile industry, clothing industry and chemical fiber manufacturing industry. Based on the panel data of the textile and apparel industry from 2010 to 2019, this paper measures green total factor productivity (GTFP) by using the unexpected output super efficiency SBM model and the ML index. On this basis, this paper empirically tests the impact of digital economy on the GTFP of textile and apparel industry, and the dual intermediary effects of rationalization of industrial structure and advanced industrial structure are discussed. The results show that: (1) The GTFP of the textile and apparel industry shows a fluctuating upward trend, but it is in a state of low growth. (2) Digital economy has a significant effect on promoting the GTFP. Among them, it has a positive effect on the improvement of GTFP in textile industry, but has no obvious effect on the clothing industry, and has a restraining effect on the chemical fiber manufacturing industry. (3) In the process of the impact of digital economy on GTFP, the rationalization of industrial structure has a partial intermediary effect, and the level of effect reaches 35.81%, while the advancement of industrial structure does not necessarily have a "structural dividend", and its influence on GTFP is non-linear. This paper enriches the research on the influencing factors of GTFP, and is also an effective supplement to the research on digital economy. The conclusions provide a reliable empirical basis for digital economy to help the textile and apparel industry pollution control, and also provide policy references for giving full play to the green value of digital economy.

## Introduction

Textile and apparel industry is a basic industry that solves people's livelihood problems and promotes national economic and social development [1]. In 2019, China's textile and apparel industry had good economic benefits, with the GDP of industrial enterprises above designated

**Data Availability Statement:** All relevant data are within the manuscript and its Supporting Information files.

**Funding:** The authors gratefully acknowledge the financial support provided by Shanxi Federation of Social Sciences (Nos. KY2021141,http://www.sxskw.org.cn/), and Taiyuan Development and Reform Commission (2021.9-2021.12,http://fgw.taiyuan.gov.cn/). The funders had role in decision to publish and preparation of the manuscript.

**Competing interests:** The authors have declared that no competing interests exist.

size increasing by 6.9% year on year. However, the use of energy is still growing rapidly, and the problem of greenhouse gas emissions are still serious [2]. According to the report of the United Nations Environment Programme (UNEP), the textile and apparel industry has become the second largest polluting industry after the oil industry. Consuming 7,500L of water for making each pair of jeans, is equivalent to one person's water for seven years. Wastewater discharge accounts for 20% of the world's waste water. Greenhouse gas emissions have far exceeded the total emissions of flights and shipping. According to the second National General Survey of pollution sources, China's textile and apparel industry ranks among the top three industrial sources in terms of ammonia nitrogen emissions, total nitrogen emissions, chemical oxygen demand and so on. Therefore, Textile Industry Federation in 2021 emphasized that it is necessary to promote the development of "technology, green, and fashion" in the industry, and proposed to support the construction of a number of key projects such as cleaner production, pollution control and resource recycling, which requires the textile and apparel industry to continue to explore the road of emission reduction, so as to achieve the "30·60 goal". The core of the green development of the textile and apparel industry is to improve green total factor productivity (GTFP), which largely depends on the improvement of technical efficiency and technological progress [3].

The in-depth application of information technology is not only the only way to solve the resource scarcity [4], but also an important way to improve the ecological environment [5] and enhance GTFP. According to the Ministry of Industry and Information Technology, there were 41 enterprises were known as green factories in 2018. These enterprises have applied automation, informatization and intelligence throughout the whole life cycle of production and manufacturing, achieving efficient utilization of raw materials and waste recycling. Many clothing enterprises such as Taiping Bird and Hong Dou, have completed carbon footprint accounting using digital technology. It can be seen that the digital economy has become a new driving force for the sustainable development of the textile and apparel industry, but it does not match the lagging theory. How to drive the digital economy to play green value is a hot topic worthy of attention by relevant departments and academic circles. In addition, textile and apparel industry has the longest industrial chain. The differences of subdivided industries, such as the nature, development mode, development stage, and policy environment, may affect the green role of the digital economy. In order to accurately grasp the mechanism of the digital economy on the GTFP of the textile and apparel industry, it is necessary to analyze the heterogeneity of the digital economy to the sub-industry GTFP from the perspective of the whole industry chain. Secondly, digital technology is a new element in addition to the traditional factors of production. The production method formed based on data assets will drive the transformation and upgrading of traditional industries [6]. As an important link between factors of production and the environment [7], industrial structure plays the role of "resource converter" and "pollutant controller" [8], contributing more than 70% to the reduction of energy consumption. Therefore, investigating the mediating role of industrial structure can provide new theoretical insights for the development of GTFP driven by digital economy.

Based on the above analysis, this paper proposes a theoretical model for the promotion of GTFP driven by the digital economy under the intermediary effect of industrial structure upgrading, and proposes the following research questions: (1) Can digital economy significantly improve the GTFP of the textile and apparel industry? Considering the segmentation differences within the industry, is there heterogeneity in the impact of the digital economy on GTFP in different segments of the textile and apparel industry? (2) Does the upgrading of industrial structure play an intermediary role in the promotion of GTFP driven by the digital economy, and if so, how it is affected?

In view of this, this paper first calculates the GTFP of textile and apparel industry by using super-efficiency SBM model and ML index, then empirically tests the impact mechanism of the digital economy on GTFP, and further measures and compares the heterogeneous impact of the digital economy on the GTFP of the three sub-sectors. Finally, the mediation effect is used to explore the relationship between digital economy, the upgrading of industrial structure and GTFP. These conclusions have important theoretical and practical significance for giving full play to the green value of the digital economy and making the textile and apparel industry walk out of the green path.

## Literature review

### Research on GTFP

Many scholars believe that total factor productivity (TFP) does not take energy and ecological environment into account, which challenges the accuracy of productivity measurement. They pointed out that GTFP is in line with the guidance of contemporary green development [9]. GTFP evaluation began in Occident. The definition of GTFP is: on the basis of TFP, energy consumption is used as an input factor, and pollution emission is introduced into the production function as an undesired output, and the TFP obtained is GTFP [10], which is used to measure the green development performance and high-quality development level of an economy [11]. From the perspective of GTFP measurement, there are two main measurement methods, namely parametric method and non-parametric method. The parameter method needs to construct a "multi -input and single- output" production model in advance. The non-parametric method does not need to build a function, which has a wider application scope [12,13], mainly including CCR, BCC, FG, ST and SBM models. Compared with other models, the SBM model not only has the characteristics of non-radial and non-angular, which will not be overestimated or underestimate the level of productivity, but also can effectively solve the problem of efficiency evaluation when there is undesired output, becoming an important model for GTFP research. Since the SBM model measures the static efficiency, Chung et al. creatively proposed the ML (Malmquist-Luenberger) index [14], which can not only explore the dynamic growth of GTFP, but also further clarify the source of GTFP growth, making GTFP measurement more comprehensive and accurately [15,16]. The combination of the two models has advantages in evaluating the development level of GTFP and the variation of GTFP, which has been widely used in efficiency evaluation of all walks of life, and also become the method for measuring GTFP of the textile and apparel industry in this paper. From the perspective of research objects and scales, researches of GTFP from a single province to an economic region to the whole country is involved. At the same time, researches cover industries such as agriculture [17], electric power [18], hotel [19] and other industries. However, as a key industry of energy conservation, emission reduction and consumption reduction, GTFP of textile and apparel industry has not received due attention.

### Research on digital economy

At present, the research on the digital economy is still in initial stage. The impact of the digital economy on the macro-economy, meso-industry and micro-enterprises has become a recent research hotspot. From the micro perspective, the development of the digital economy can significantly improve the efficiency of resource allocation [20] and innovation capabilities of enterprises [21]. From the medium perspective, the value of the digital economy is mainly reflected in the promotion of industrial production efficiency, operation innovation and industrial structural transformation [22]. From the macro perspective, existing studies have examined the welfare effects of the digital economy from the aspects of digital trade [23],

digital financial inclusion [24], and carbon emissions [25], etc. However, there are still few studies on the green effects of digital economy.

In recent years, the measurement of the digital economy has attracted attention. Some institutions and scholars have discussed the measurement indicators of the digital economy. For example, The UK Bureau of Statistics and the US Department of Commerce's Bureau of Economic Analysis believe that the digital economy includes e-commerce and infrastructure. Russia defines the digital economy as ICT industry and digital content. China Financial Think Tank evaluates the development level of industrial digitalization from the perspective of digital industrialization and industrial digitalization. The China Academy of Information and Communications believes that the digital economy development index should be composed of digital industrialization, industrial digitalization, digital governance, and data value. Li and Han constructed three-dimensional indicators of digital infrastructure, digital industrialization, and industrial digitization [26]. Wang et al. constructed the index system from the dimensions of infrastructure, digital industrialization, industrial digitization and digital economy development environment [27]. It can be seen that digital infrastructure, digital industrialization and industrial digitization are currently generally recognized indicators, so they have also become the indicators for measuring the development level of the digital economy in this paper.

## Research on the relationship between the digital economy and GTFP

Existing studies have carried out research on the relationship between information technology and environmental quality under the background of the Internet, intelligent manufacturing, and informatization [28], but there is a lot of controversy [29]. Some scholars believe that the development of information technology will accelerate the development of green economy [30]. For example, Shabani and Shahnazi found that information technology itself is environmentally friendly, and the carbon dioxide emissions of this industry are far lower than those of other sectors [31], which suggests that the rise of the Internet is good for the green economy. Moyer and Hughesbelieved that the development of the information industry can achieve green development by improving production efficiency, reducing energy intensity and increasing the utilization rate of renewable energy [32]. However, some scholars believe that information technology has a negative effect on the ecological environment. Bernstein and Madlener, taking specific industries in Europe as examples, found that informatization has brought about an increase in power consumption [33]. The rapid development of information technology will increase people's looting of natural resources [34]. Fan and Li pointed out that measuring the digital economy with information technology can't fully reflect the form of the digital economy [35], which inspires scholars to further explore the green effect of the digital economy.

Cheng and Qian took the lead in using empirical methods to conclude that the impact of the digital economy on industrial GTFP has a threshold effect of industry scale and institutional environment [36]; Wu et al. later found that the digital economy is affected by a single threshold effect of R&D investment in the process of affecting GTFP [37]. The above research reveals that the impact of the digital economy on GTFP exists and is non-linear. However, the threshold regression model is used to only examine the coordination and adaptation of variables, and does not reveal the internal impact mechanism. Therefore, these articles inspire to find that there may be a mediating effect when thinking about the relationship between the digital economy and GTFP. Another part of scholars draws on the points of information technology and ecological environment, and believes that the digital economy also has certain negative effects on green development. The technological progress brought by the digital economy will enable relevant enterprises to increase output by resetting or updating equipment,

developing new materials, and strengthening energy consumption [25]. Only by optimizing the allocation of resource elements, strengthening the connection between industrial chains, and leading the rationalization and upgrading of the industrial structure, can it contribute to carbon emission reduction and energy efficiency improvement [38], and truly give full play to the green value that the digital economy should have. Therefore, industrial structure is likely to be an intermediary variable for digital economy to promote industrial GTFP.

To sum up, the existing researches have carried out some useful exploration of GTFP and digital economy, providing the basis for the selection of indicators, method application and model construction, but there is still room for further research. First, few studies focus on the green development of the textile and apparel industry. Considering that the textile and apparel industry has become the second largest polluting industry in the world, the research on GTFP measurement and influencing factors is in line with the development concept of "building a beautiful China and weaving beautiful clothes", and it is also of practical significance to improve the green development level of the textile and apparel industry. Second, the studies on GTFP driven by the digital economy are mainly qualitative analysis, lacking scientific quantitative evaluation. Only a few literatures use the threshold model to examine the adaptation problem between variables, and the internal mechanism has not been revealed.

The contribution of this study mainly has the following three points: Firstly, the research on the development of textile and apparel industry takes into account total factor productivity and does not include energy input and non-expected output, which leads to the error of over-estimating the benefits of industry. This paper pays attention to the intensive development of textile and apparel industry, and calculates the green efficiency of textile and apparel industry and subdivided industry. Secondly, there is a lack of literature on the impact of digital economy on green development. This paper makes a useful exploration of the relationship between digitalization and greening of the textile and apparel industry from both theoretical and empirical perspectives, which enriches the research on green development and pollution control, and is also an effective supplement to the related research on digital economy. Thirdly, the upgrading of industrial structure is introduced as an intermediary variable, and the mediating effect model is used to reveal the internal mechanism of the influence of digital economy on GTFP, providing a new theoretical insight for the development of GTFP driven by digital economy.

## Theoretical analysis and research hypotheses

### The direct mechanism of digital economy on the improvement of GTFP in the textile and apparel industry

It is widely believed in the academic circle that the digital economy covers digital infrastructure, digital industrialization and industrial digitization. Digital industrialization is manifested as digital products with data as production factors that can be traded on the market through data sorting and analysis, and realizes new industrial forms of information value-added, including telecom service industry, internet industry, software and information service industry, etc. Digital industry itself has the characteristics of environmental friendliness, enabling the textile and apparel industry to improve its GTFP by constantly providing new technologies, products and services [39]. Industrial digitalization can carry out all-round, multi-angle and whole- chain digital transformation of traditional industries, reducing the dependence of traditional industries on resources. It can promote green innovation and efficiency improvement of industries by achieving technological breakthroughs in decarbonization and zero carbon through intelligent manufacturing [40]. Meanwhile, with the implementation of the "new infrastructure" strategy, digital infrastructure has gradually become an important aspect of the

digital economy. On the one hand, the improvement of intelligent manufacturing infrastructure can be used to monitor pollution emissions in real time, and provide quantitative parameter standards for industrial carbon emissions [41]. On the other hand, digital media can be used to convey the value of green consumption to the public, and guide the public to achieve green consumption. Based on this, the following hypotheses are proposed:

Hypothesis 1: Digital economy has a positive effect on the improvement of GTFP.

## The indirect mechanism of digital economy on the promotion of GTFP in the textile and apparel industry

The upgrading of industrial structure generally includes two dimensions: rationalization of industrial structure and advanced industrial structure [42,43] The rationalization of industrial structure is a process of obtaining good benefits by coordinating the relationship between various departments in the industry on the basis of the unchanged existing resources. Advanced industrial structure is the transformation process of industrial structure from low-level (labor intensive) to high—level (technology and knowledge intensive), and from traditional high emission industry to low carbon industry. The indirect effects of the upgrading of the industrial structure are as follows: First, the rationalization of the industrial structure helps to enhance the impact of the digital economy on the industrial GTFP. Digital economy can accurately connect demand and supply, break down the administrative barriers to the flow of factors, and improve the system coordination ability among industries, and then promote the rationalization of the industrial structure. The positive effect of industrial structure rationalization on the promotion of GTFP has been confirmed in many studies [44]. The rationalization of industrial structure can bring about a substantial increase in the efficiency of resource use [45], thereby greatly reducing energy consumption and promoting the development of GTFP. Second, advanced industrial structure helps to enhance the influence of digital economy on industrial GTFP. Affected by the long-term extensive development mode, China's textile and apparel industry is still a labor-intensive industry with low added value. New forms of business derived from digital economy, such as intelligent manufacturing, platform economy and personalized service, can strengthen market competition mechanism and eliminate polluting enterprises, thereby promoting the transformation of textile and apparel industry 's structure from high-emission to environment-friendly. At the same time, industrial robots are used to complete relevant work [46], so as to realize the advanced industrial structure, and finally achieve the synchronous development of economic and ecological values. Therefore, the following hypothesis is proposed:

Hypothesis 2: Digital economy plays a positive role in promoting the upgrading of industrial structure.

Hypothesis 3: The upgrading of industrial structure has a positive effect on the promotion of GTFP.

Hypothesis 4: Industrial structure upgrading plays an intermediary role in the process of digital economy promoting GTFP.

In summary, this paper presents a theoretical framework diagram of the impact of the digital economy on GTFP (Fig 1).

## Empirical model building

### Variable selection

**Explained variable.**   Compared with the traditional DEA model, the super-efficiency SBM model allows the efficiency values to be greater than 1, which can be used for comparative

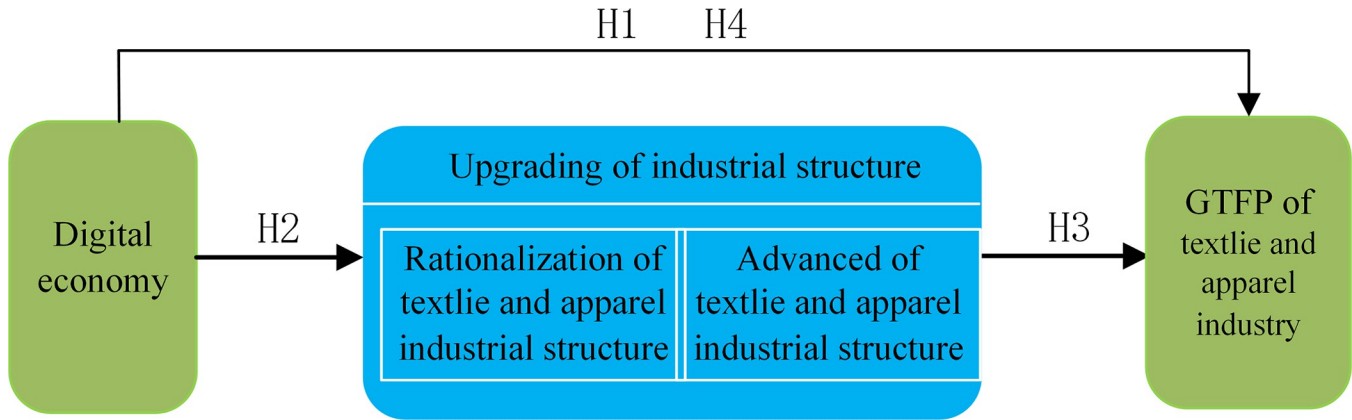

**Fig 1. Theoretical framework for the impact of the digital economy on GTFP.**

evaluation of effective units. At the same time, the super-efficiency SBM effectively overcomes the relaxation problems in the input and output variables, and solves the efficiency evaluation problems of non-expected output in the output factors. It has been widely used in the evaluation of ecological efficiency and environmental efficiency. This paper uses this model to evaluate the GTFP of the textile and apparel industry. The basic model is:

$$\theta = min \frac{1 - \frac{1}{N}\sum_{n=1}^{N}\frac{s_n^x}{x_{kn}^t}}{1 + \frac{1}{M+I}\left(\sum_{m=1}^{M}\frac{s_m^y}{y_{km}^t} + \sum_{i=1}^{I}\frac{s_i^b}{b_{ki}^t}\right)} \tag{1}$$

$$\begin{cases} x_{kn}^t = \sum_{t=1}^{T}\sum_{k=1}^{K}\lambda_k^t x_{kn}^t + s_n^x, n = 1, \cdots, N \\ y_{km}^t = \sum_{t=1}^{T}\sum_{k=1}^{K}\lambda_k^t y_{km}^t - s_m^y, m = 1, \cdots, M \\ b_{ki}^t = \sum_{t=1}^{T}\sum_{k=1}^{K}\lambda_k^t b_{ki}^t + s_i^b, i = 1, \cdots, I \\ \lambda_k^t \geq 0, s_n^x \geq 0, s_m^y \geq 0, s_i^b \geq 0, k = 1, \cdots, K \end{cases}$$

Where, $t$, $k$ represent the year and region respectively; $N$, $M$ and $I$ are the number of input indicators, expected output indicators, and undesired output indicators, respectively; $x$ is input index; $y$ is expected output; $b$ is the undesired output; $s_n^x, s_m^y, s_i^b$ represent the n-th input slack variable, the m-th expected output slack variable and the i-th undesired output slack variable, respectively; $x_{kn}^t, y_{km}^t, b_{ki}^t$ are the input, expected output and undesired output of region $k$ in period $t$, respectively; $\lambda$ is the weight coefficient; $\theta$ is the efficiency value. If $\theta \geq 1$, it means that the GTFP of textile and apparel industry is valid, and if $\theta < 1$, the production is invalid.

Since super-efficiency SBM model can only measure static efficiency and only reflect the relationship between each unit and the frontier, but cannot reflect the position change of production unit and frontier. Therefore, this paper studies the trend characteristics of GTFP growth with the help of Malmquist-Luenberger (ML) index. It is decomposed into technical efficiency improvement index (EC) and technological progress index (TC). Referring to the

study of Chung et al. [14], ML index from $t$ period to $t$+1 period is:

$$ML_t^{t+1} = \left\{ \frac{[1 + D_0^t(x^t, y^t, b^t, g^t)]}{[1 + D_0^t(x^{t+1}, y^{t+1}, b^{t+1}, g^{t+1})]} \right.$$

$$\left. \times \frac{[1 + D_0^{t+1}(x^t, y^t, b^t, g^t)]}{[1 + D_0^{t+1}(x^{t+1}, y^{t+1}, b^{t+1}, g^{t+1})]} \right\}^{\frac{1}{2}} \tag{2}$$

The specific process of ML index decomposition is as follows:

$$ML_t^{t+1} = EC_t^{t+1} \times TC_t^{t+1} = \left[ \frac{1 + D_0^t(x^t, y^t, b^t, g^t)}{1 + D_0^{t+1}(x^{t+1}, y^{t+1}, b^{t+1}, g^{t+1})} \right]$$

$$\times \left[ \frac{1 + D_0^{t+1}(x^t, y^t, b^t, g^t)}{1 + D_0^t(x^t, y^t, b^t, g^t)} \times \frac{1 + D_0^{t+1}(x^{t+1}, y^{t+1}, b^{t+1}, g^{t+1})}{1 + D_0^t(x^{t+1}, y^{t+1}, b^{t+1}, g^{t+1})} \right]^{\frac{1}{2}} \tag{3}$$

Where, $x$ is the input factor; $y$ is expected output; $b$ is the undesired output; $g$ is the slack variable; $D_0^t(x^t, y^t, b^t, g^t)$, $D_0^{t+1}(x^{t+1}, y^{t+1}, b^{t+1}, g^{t+1})$ represent the distance function of the $t$ and $t$+1 periods; $D_0^t(x^{t+1}, y^{t+1}, b^{t+1}, g^{t+1})$ represents the mixed distance function of $t$+1 period based on $t$ period; $D_0^{t+1}(x^t, y^t, b^t, g^t)$ represents the mixed distance function of $t$ period based on $t$+1 period. If $ML$, $EC$ and $TC > (<) 1$, it indicates that green total factor productivity of textile and apparel industry increases (decreases), technical efficiency improves (deteriorates), and technological progress (retrograde).

This paper selects the input-output index of Table 1 for calculation.

**Core explanatory variable.** According to the literature review, digital infrastructure level, digital industrialization and industrial digitalization are used to measure the digital economy. The index data are all obtained from the China Statistical Yearbook, and some missing data are supplemented based on the Statistical Bulletin of National Economic and Social

**Table 1. Input-output variables of Green total factor productivity.**

| Criterion layer | Index | Measure | The data source |
|---|---|---|---|
| Input | Land | Number of enterprises in textile and apparel industry and subdivision industry above designated size, unit: pieces | China Statistical Yearbook |
| | Capital | Capital stock of textile and apparel industry and subdivision industry above designated size, unit:100 million yuan | Refer to the algorithms of Hall and Jones [47]: $K_{i,t} = K_{i,t-1}(1 - \delta_{i,t}) + I_{i,t}$. Among them, $i$ is subdivided industry, $t$ is time, $K$ is capital stock, $I$ is amount of investment in fixed asset investment. $\delta_{i,t}$ is the depreciation rate of i industry in $t$ years. |
| | Labor | Number of persons in textile and apparel industry and subdivided industry above designated size, unit: ten thousand people | China Statistical Yearbook |
| | Energy consumption | Six kinds of energy consumption (10,000 tons of standard coal) | According to the China Statistical Yearbook, the textile and apparel industry mainly consumes coal, gasoline, kerosene, diesel, fuel oil and electricity. |
| Output | Expect output | Main business income of textile and apparel industry and sub-industry above designated size, unit: 100 million yuan | China Statistical Yearbook |
| | Unexpected output | Carbon emissions of textile and apparel industries and sub-industries above designated size, unit: 10,000 tons of standard coal | $C = \sum_i^6 E_i \times f_i \times k_i$. Where $C$ is the carbon emission, $E_i$ is the energy consumption of the $i$ energy. $f_i$ is the discount coal coefficient of the $i$ energy, $k_i$ is the carbon emission coefficient of the $i$ energy. |

**Table 2. Digital economy evaluation index system.**

| First level indicators | Weight | Description | Weight |
|---|---|---|---|
| Digital Infrastructure Index | 0.163 | Length of long-distance optical fiber lines, unit: ten thousand km | 0.005 |
| | | Number of Internet access ports, unit: ten thousand pieces | 0.141 |
| | | Capacity of mobile telephone exchange, unit: ten thousand households | 0.018 |
| Digital Industrialization Index | 0.464 | Internet penetration, unit: % | 0.020 |
| | | The mobile phone penetration, unit: % | 0.019 |
| | | The fixed telephone penetration, unit: % | 0.016 |
| | | Total telecom business, unit: hundred million | 0.285 |
| | | Software business income, unit: ten thousand | 0.124 |
| Industry Digitization Index | 0.372 | Number of computers used per 100 people, unit: units | 0.025 |
| | | Proportion of enterprises with E-commerce transactions, unit: % | 0.133 |
| | | Volume of E-commerce sales, unit: hundred million | 0.214 |

Development and Wind Database. Since the dimensions of indicators are different, the entropy weight method is used to calculate the weight of indicators, and the results are shown in Table 2.

Then, the comprehensive index and sub-index of digital economy development are calculated, and the results are shown in Table 3.

$$I_j = \frac{\sum X'_{ij} w_i}{\sum w_i} \quad (4)$$

where, $X'_{ij}$ is the value after the treatment of initial valuation, $w_i$ is the weight of indicator $i$; $I_j$ is the digital economy comprehensive development index, digital infrastructure index, digital industrialization index and industrial digitization index in the year j.

**Metavariable variable.** The rationalization of industrial structure is a measure of the coupling degree of input and output, which reflects the coordination between industries and the allocation of resources. Most studies choose structural deviation degree to measure the rationalization of industrial structure, which cannot reflect the importance of each industry. Theil Index can make up for this deficiency and retain the economic significance of structural deviation degree. Theil index is expressed as:

$$TL = \sum_{i=1}^{n} \frac{Y_i}{Y} ln \left( \frac{\frac{Y_i}{Y}}{\frac{L_i}{L}} \right) i = 1, 2, 3 \quad (5)$$

**Table 3. Estimation of the digital economy development level from 2010–2019.**

| Year | Digital economy comprehensive development index | Digital infrastructure index | Digital industrialization Index | Industrial digitization index |
|---|---|---|---|---|
| 2010 | 1.000 | 1.000 | 1.000 | 1.000 |
| 2011 | 1.209 | 1.221 | 0.738 | 1.792 |
| 2012 | 1.554 | 1.637 | 0.889 | 2.346 |
| 2013 | 2.083 | 1.822 | 1.066 | 3.467 |
| 2014 | 2.709 | 2.040 | 1.245 | 4.831 |
| 2015 | 3.280 | 2.837 | 1.465 | 5.741 |
| 2016 | 3.702 | 3.460 | 1.415 | 6.661 |
| 2017 | 4.259 | 3.768 | 1.800 | 7.544 |
| 2018 | 5.169 | 4.198 | 2.722 | 8.649 |
| 2019 | 5.726 | 4.432 | 3.772 | 8.734 |

where $Y$ is the total income of textile and apparel industry; $Y_i$ is the income of each subdivided industry; $L$ is the total labor force of textile and apparel industry; $L_i$ is the number of labor force in each subdivided industry. The smaller the $TL$ is, the better the coordination of each subdivision industry is, that is, the more reasonable the industrial structure.

The essence of advanced industrial structure is a process of orderly evolution of industrial structure from low-level to higher-level with the infiltration of highly skilled labor force and emerging technology. The formula can be expressed as follows:

$$TS_{it} = \sum_{i=1}^{3} y_{it} \times lp_{it} \tag{6}$$

$$lp_{it} = \frac{Y_{it}}{L_{it}} \tag{7}$$

Where $y_{it}$ is the proportion of industrial added value in GDP of $i$ industry in $t$ period; $lp_{it}$ is the labor rate of $i$ industry in $t$ period; $Y_{it}$ is the added value of $i$ industry in $t$ period; $L_{it}$ is the employment of $i$ industry in $t$ period. The greater the value of $TS$, the better the advanced level of each subdivision industry.

**Control variables.** With reference to the relevant studies of GTFP and considering the availability of data, the control variables selected were foreign direct investment (FDI), education level (EDU), and policy environment (PL). FDI is measured by the amount of foreign capital used, converted into RMB at the exchange rate of the current year. EDU is measured by the proportion of education expenditure in the general public budget. PL is measured by government expenditure on science and technology.

**Descriptive statistics.** Considering the possible heteroscedasticity between variables, we log the variables, and descriptive statistics are given in Table 4.

## Model setting

**Benchmark model.** In order to investigate the impact of digital economy on GTFP, this paper constructed the following panel model:

$$lnGTFP_{it} = \alpha_0 + \alpha_1 lnDIGE + \alpha_2 lnCV_{it} + \phi_i + v_t + \varepsilon_{it} \tag{8}$$

In the Eq 8, $GTFP_{it}$ represents the green total factor productivity of $i$ industry in $t$ year; $DIGE$ is the development level of digital economy; $CV$ is the collection of the control variables; $\alpha_1$ is the total effect of digital economy on GTFP development; $\phi_i$ represents the fixed effect of industry; $v_t$ represents the fixed effect of time; $\varepsilon_{it}$ represents the random error term.

**Mediation effect.** In order to test the mechanism of industrial structure rationalization and advanced industrial structure, this paper uses three-step gradual regression method for

**Table 4. Descriptive statistics of variables.**

|  | Variables | Mean | Std. | Min | Max |
|---|---|---|---|---|---|
| Explained variable | GTFP | 1.220 | 0.429 | 0.239 | 1.827 |
| Explanatory variables | ln DIGE | 0.969 | 0.585 | 0.000 | 1.745 |
| Metavariable | ln TL | 0.009 | 0.041 | -0.116 | 0.078 |
|  | ln TS | 5.493 | 3.654 | 1.132 | 14.59 |
| Control variables | ln FDI | 9.047 | 0.074 | 8.924 | 9.162 |
|  | EDU | 0.151 | 0.008 | 0.140 | 0.169 |
|  | ln PL | 8.639 | 0.326 | 8.086 | 9.156 |

**Table 5. SBM value of Super efficiency of textile and apparel industry from 2010 to 2019.**

| Year | 2010 | 2011 | 2012 | 2013 | 2014 | 2015 | 2016 | 2017 | 2018 | 2019 | Mean |
|---|---|---|---|---|---|---|---|---|---|---|---|
| Textile industry | 0.272 | 0.239 | 0.768 | 1.044 | 1.079 | 1.064 | 1.054 | 1.043 | 1.084 | 1.024 | 0.861 |
| Clothing industry | 1.521 | 1.528 | 1.606 | 1.631 | 1.613 | 1.792 | 1.807 | 1.827 | 1.756 | 1.757 | 1.653 |
| Chemical fiber industry | 1.733 | 1.725 | 1.408 | 1.349 | 1.324 | 1.308 | 1.331 | 1.376 | 1.517 | 1.562 | 1.463 |
| Mean | 1.072 | 1.164 | 1.261 | 1.341 | 1.339 | 1.388 | 1.397 | 1.415 | 1.452 | 1.448 | 1.328 |

verification. The first step is to test whether digital economy plays a significant role in the industrial structure upgrading:

$$lnTL_{it} = \beta_{01} + \beta_{11}lnDIGE + \beta_{21}lnCV_{it} + \phi_i + \nu_t + \varepsilon_{it} \tag{9}$$

$$lnTS_{it} = \beta_{02} + \beta_{12}lnDIGE + \beta_{22}lnCV_{it} + \phi_i + \nu_t + \varepsilon_{it} \tag{10}$$

The second step is to add digital economy and intermediary variables into Eq 9 and Eq 10:

$$lnGTFP_{it} = \chi_{01} + \chi_{11}lnDIGE + \chi_{21}lnTL_{it} + \chi_{31}lnCV_{it} + \phi_i + \nu_t + \varepsilon_{it} \tag{11}$$

$$lnGTFP_{it} = \chi_{02} + \chi_{12}lnDIGE + \chi_{22}lnTS_{it} + \chi_{32}lnCV_{it} + \phi_i + \nu_t + \varepsilon_{it} \tag{12}$$

The third step is to distinguish between full mediation and partial mediation.

In the above formula, $TL_{it}$ and $TS_{it}$ represent the rationalization and advanced industrial structure respectively; $\beta_{11}$ and $\beta_{12}$ represent the direct effects of digital economy on the rationalization and upgrading of industrial structure respectively; $\chi_{11}$ and $\chi_{12}$ represent the direct effects of digital economy on GTFP; $\chi_{21}$ and $\chi_{22}$ represent the direct effects of rationalization of industrial structure and advanced industrial structure on GTFP respectively.

## Empirical analysis

### GTFP analysis of textile and apparel industry

**Static analysis of GTFP.** In this paper, the unexpected output super-efficiency SBM model with constant scale return is selected to measure the GTFP of textile and apparel industry from 2010 to 2019. The results are shown in Table 5 and Fig 2. Overall, the average value of GTFP in the textile and apparel industry from 2010 to 2019 was 1.328, exceeding 32.8% of the optimal level of GTFP, indicating that the textile and apparel industry is showing a green and healthy development trend. In terms of different industries, the GTFP of the three subdivided industries changes are wavy, but it has generally shown an upward trend in recent years, mainly due to the in-depth implementation of the concept of green economy development. GTFP in the clothing industry is showing the leading trend (The average value is 1.653), and its average efficiency level has been at the forefront of production since 2010. GTFP in the

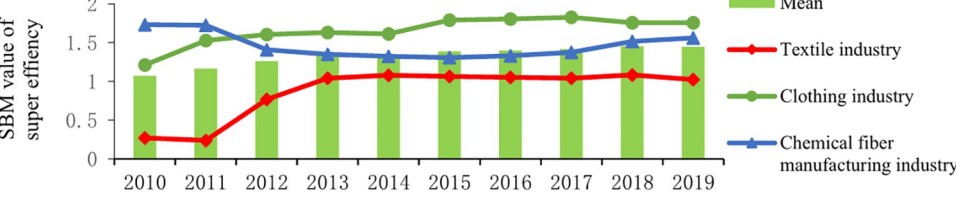

**Fig 2. SBM value of super efficiency of textile and apparel industry from 2010 to 2019.**

**Table 6. Decomposition of ML of GTFP in textile and apparel industry during 2010–2019.**

| Year | Textile industry | | | Clothing industry | | | Chemical fiber industry | | |
|------|------|------|------|------|------|------|------|------|------|
| | EC | TC | ML | EC | TC | ML | EC | TC | ML |
| 2010–2011 | 0.883 | 1.202 | 1.062 | 1.005 | 1.023 | 1.028 | 0.996 | 1.071 | 1.066 |
| 2011–2012 | 0.996 | 1.071 | 1.511 | 1.051 | 1.831 | 1.924 | 0.816 | 1.044 | 0.851 |
| 2012–2013 | 1.359 | 0.981 | 1.334 | 1.016 | 0.987 | 1.003 | 0.959 | 0.980 | 0.940 |
| 2013–2014 | 1.034 | 1.012 | 1.046 | 0.988 | 1.034 | 1.022 | 0.982 | 1.020 | 1.002 |
| 2014–2015 | 0.986 | 0.844 | 0.832 | 1.111 | 0.972 | 1.080 | 0.987 | 0.995 | 0.983 |
| 2015–2016 | 0.991 | 1.022 | 1.013 | 1.008 | 1.145 | 1.154 | 1.078 | 0.999 | 1.017 |
| 2016–2017 | 0.990 | 0.989 | 0.980 | 1.011 | 1.017 | 1.029 | 1.034 | 0.991 | 1.023 |
| 2017–2018 | 1.039 | 0.990 | 1.029 | 0.961 | 1.075 | 1.032 | 1.103 | 0.967 | 1.067 |
| 2018–2019 | 0.945 | 0.908 | 0.858 | 1.001 | 0.986 | 0.987 | 1.029 | 1.072 | 1.103 |
| Mean | 1.048 | 1.023 | 1.074 | 1.017 | 1.119 | 1.140 | 0.991 | 1.015 | 1.006 |

chemical fiber manufacturing industry remained at a moderate level (The average value is 1.463). After a decline in efficiency from 2010 to 2015, GTFP was improved from 2016 to 2019, due to the green fiber label registration of the Chemical Fiber Industry Institute in recent years. The efficiency value of the textile industry is at the lowest level (The average is 0.861). Although it increased significantly from 2011, the average did not reach the forefront of the production level, which shows that the resource utilization of the industry is not high, and the carbon emissions have not been effectively controlled. Therefore, the textile industry should focus on strengthening low-carbon management.

**Dynamic analysis of GTFP.** In order to further investigate the factors influencing the growth of GTFP and the change rate of efficiency, the ML index was used to further measure the change rate of GTFP. The results are shown in Table 6. It can be seen the ML index of GTFP of the three subdivided industries from 2010 to 2019 was mostly greater than 1, indicating that the GTFP of the textile and apparel industry is growing. Moreover, the ML value from largest to smallest is as follows: clothing industry> textile industry> chemical fiber industry. Combined with the static analysis results, it can be concluded that the clothing industry is characterized by high efficiency (1.653) and high growth (1.140), the chemical fiber industry shows the characteristics of medium efficiency (1.463) and low growth (1.006), and the textile industry shows the characteristics of low efficiency (0.861) and medium growth (1.074).

In order to explore the influencing factors of GTFP change, the ML index was decomposed into technical efficiency (EC)and technological progress (TC). From the perspective of growth sources, the average of EC and TC of textile and clothing industry is greater than 1. It shows that technical efficiency and technological progress jointly promote the improvement of GTFP in the two industries. The technical progress index of chemical fiber industry is effective, but the average of technical efficiency index is smaller than 1, indicating that the industrial operation and management ability can't timely follow up, and the allocation of factors is unreasonable.

## Benchmark regression analysis

Before the model regression, the unit root and co-integration test on the panel data were first performed. The IPS and Fisher-ADF tests found that after the first-order difference conversion, variables passed the significant test, so they were first-order differential stable sequences, and the co-integration test can be performed. The Pedroni and Kao tests showed that the null hypothesis was rejected, indicating that there was a long-term co-integration relationship between variables, and there will be no spurious regression. Using the F test and Hausman

**Table 7. Regression results of digital economy on GTFP.**

| Variables | Model (1) | Model (2) | Model (3) | Model (4) |
|---|---|---|---|---|
| DIGE | 0.182*** | 0.368*** | 0.363*** | 0.389*** |
| | (9.185) | (7.323) | (8.377) | (4.833) |
| FDI | | -1.392*** | -1.283*** | -1.203*** |
| | | (-3.492) | (-3.646) | (-2.617) |
| EDU | | | 2.535** | 2.787** |
| | | | (2.368) | (2.276) |
| PL | | | | -0.0606 |
| | | | | (-0.327) |
| Constant | 1.046*** | 13.46*** | 12.09*** | 11.83*** |
| | (46.73) | (3.780) | (3.781) | (3.459) |
| N | 40 | 40 | 40 | 40 |

Notes: The t-statistic in parenthesis.

***, **and * represent significant levels of 1%, 5% and 10%. The same below.

test, the fixed effect model is the most suitable for benchmark regression. On this basis, the FGLS model was used to avoid certain heteroscedasticity and sequence correlation. At the same time, to avoid multicollinearity, stepwise regression method was used. The specific results are shown in Table 7. The Model (1) in Table 7 did not add any control variables, and model (2)—(4) showed the regression results after adding control variables one by one.

According to the regression coefficient of core explanatory variables, digital economy has a significant positive impact on the industrial upgrading of textile and apparel industry. Every unit increase in the development level of digital economy can promote the industrial upgrading of textile and apparel industry by 0.389 units, indicating that digital economy has the green value. Digital economy can grow into a key driving force for improving industrial production efficiency under the background of "clear water and green mountains are gold and silver mountains", which is also consistent with Cheng and Qian's conclusion [36], and H1 has been verified. From the perspective of control variables, foreign direct investment has a significant negative impact in the model. The reason may be that the introduction of foreign investment is easy to form technology dependence, which is not conducive to the innovation and high-quality development of the textile and apparel industry. The development of human capital has a significant positive impact on the GTFP of the textile and apparel industry, indicating that the training of high-quality talents is conducive to the improvement of GTFP.

### Benchmark regression of digital economy to subdivided industries

Due to industry differences in the development of the digital economy, the boosting effect of digital economy on the improvement of GTFP may vary to some extent. Therefore, this paper adopts the fixed -effect model to test industry heterogeneity. The specific results are shown in Table 8. As can be seen, the influence of digital economy on GTFP of textile industry is significantly positive at 5% level. The positive promotion effect of digital economy on the clothing industry is not significant, which may be due to the high cost of digital management and the lack of relevant preferential policies. Considering the factors of high cost and high risk, enterprises are not enthusiastic about digital investment, which to some extent inhibits the play of the green function of digital economy. Digital economy has a significant negative effect on the chemical fiber manufacturing industry, which may be due to the fact that the chemical fiber manufacturing industry belongs to the heavy chemical industry [48], and its production is

**Table 8. Results of heterogeneity analysis in textile and apparel industry.**

| Variables | Textile industry | Clothing industry | Chemical fiber manufacturing industry |
|-----------|------------------|-------------------|----------------------------------------|
| DIGE | 1.440** | 0.404 | -1.146*** |
|  | (3.84) | (1.83) | (-5.67) |
| FDI | -3.790 | 0.532 | 1.385 |
|  | (-1.77) | (0.42) | (1.20) |
| EDU | 12.535* | 1.871 | 13.139*** |
|  | (2.19) | (0.56) | (4.27) |
| PL | -0.883 | -0.531 | 1.485** |
|  | (-1.02) | (-1.04) | (3.19) |
| Constant | 39.495* | 0.780 | -20.798* |
|  | (2.47) | (0.08) | (-2.42) |
| $R^2$ | 0.943 | 0.837 | 0.930 |

related to cotton and oil, so the energy consumption and carbon emission intensity are high. Although digital economy can promote the industry to update the existing production mode and carry out technological innovation, there is a certain lag effect. In addition, Digital economy makes many enterprises obtain advanced management experience and generate the spillover of technology and knowledge, forming a "demonstration effect". For example, China Textile Industry Federation awarded eight enterprises as "energy saving and emission reduction demonstration enterprises", such as Tangshan Sanyou Xingda Chemical Fiber Co., Ltd., Yiwu Huading Nylon Co., Ltd., and Hangzhou Nuobang Non-Textile Co., LTD. Although the demonstration effect can expand the expected output, it will also bring more non-satisfactory output.

## Robustness test

The benchmark effect model established in this paper may be affected by the unobservable factors, resulting in biased regression results. Therefore, robustness tests are conducted from the following two perspectives to prove the validity of the conclusions of this paper. The specific results are shown in Table 9. (1) Replace core explanatory variables. This paper uses entropy weight method to calculate the development level of digital economy, so there may be some contingency. Therefore, this paper gives the same weight of digital economic indicators. After the equal weight treatment of digital economy and sub-content index, the core explanatory

**Table 9. Results of the robustness test.**

| Variables | Replace core explanatory variables | Data indent processing |
|-----------|-------------------------------------|------------------------|
| GTFP | 0.384*** | 0.380*** |
|  | (4.847) | (4.814) |
| FDI | -1.202*** | -1.144** |
|  | (-2.620) | (-2.533) |
| EDU | 2.784** | 2.821** |
|  | (2.279) | (2.346) |
| PL | -0.0578 | -0.0629 |
|  | (-0.313) | (-0.345) |
| Constant | 11.81*** | 11.32*** |
|  | (3.459) | (3.368) |
| N | 40 | 40 |

variables are still significant, so the benchmark regression in this paper has a certain robustness. (2) Data indent processing. In order to avoid the interference of outliers and ensure the integrity of the sample, this paper conducted a re-regression after data indent processing of 2.5% for all variables. It can be seen that the estimated value, sign direction and significance of GTFP by the digital economy have not changed significantly. The benchmark regression results are robust.

## The transmission mechanism of digital economy to GTFP

The above research results show that digital economy can significantly improve GTFP. Then what is the specific transmission mechanism? Theoretical analysis shows that digital economy improves GTFP through industrial structure upgrading, so the two dimensions of industrial structure (rationalization of industrial structure and advanced industrial structure) are used as intermediary variables for verification, and the regression results are shown in Table 10. In model (1), the regression coefficient ($\alpha_1$) of digital economy on GTFP passes the significance test of 1%, which is consistent with the benchmark effect result. In model (2), the regression coefficient ($\beta_{11}$) of digital economy on the rationalization of industrial structure is significantly positive and passes the 1% test. Model (3) shows the influence of digital economy and industrial structure rationalization on GTFP with entering the model at the same time. The regression coefficient ($\chi_{21}$)of industrial structure rationalization on GTFP is significantly positive at 1% level. The regression coefficient ($\chi_{11}$) of digital economy on GTFP is also significantly positive at the 1% level. Further analysis of the mediating effect shows that $\alpha_1, \chi_{21}, \beta_{11}$ and $\chi_{11}$ are all significant according to the regression results, and $\beta_{11}\chi_{21}$ and $\chi_{11}$ are the same number, indicating that digital economy can indirectly promote GTFP through rationalization of industrial structure. It can be known from $\beta_{11}\chi_{21}/\alpha_1$, the mediating effect is 35.81%, that is, in the process of the digital economy playing a role in GTFP, 35.81% is produced by the rationalization of the industrial structure.

According to the theoretical analysis, the positive effect of digital economy is conducive to the penetration of highly skilled labor force and emerging technology, deepening the advanced degree of industrial structure and promoting GTFP. The Model (1) in Table 11 shows the impact of digital economy on GTFP, which is consistent with the benchmark regression results. Model (2) shows the influence of digital economy on the upgrading of industrial

**Table 10.  Evaluation of the mediation effect of industrial structure rationalization.**

| Variables | Model (1) | Model (2) | Model (3) |
|---|---|---|---|
| | GTFP | TL | GTFP |
| TL | | | 0.995*** |
| | | | (3.00) |
| DIGE | 0.389*** | 0.140*** | 0.247*** |
| | (4.83) | (5.48) | (2.91) |
| FDI | -1.203*** | -0.179 | -0.996** |
| | (-2.62) | (-1.23) | (-2.46) |
| EDU | 2.787** | 1.085*** | 1.291 |
| | (2.28) | (2.80) | (1.13) |
| PL | -0.061 | -0.120** | 0.061 |
| | (-0.33) | (-2.05) | (0.37) |
| Constant | 11.829*** | 2.370** | 9.258*** |
| | (3.46) | (2.19) | (2.99) |
| N | 40 | 40 | 40 |

**Table 11. Evaluation of the intermediary effect of the industrial structure advanced.**

| Variables | Model (1) | Model (2) | Model (3) |
|---|---|---|---|
| | **GTFP** | **TS** | **GTFP** |
| TS | | | -0.058*** |
| | | | (-14.76) |
| DIGE | 0.389*** | 2.722*** | 0.562*** |
| | (4.83) | (4.62) | (5.29) |
| FDI | -1.203*** | 9.854*** | -0.600 |
| | (-2.62) | (2.92) | (-0.99) |
| EDU | 2.787** | 50.408*** | 5.762*** |
| | (2.28) | (5.61) | (3.54) |
| PL | -0.061 | -2.368* | -0.238 |
| | (-0.33) | (-1.74) | (-0.97) |
| Constant | 11.829*** | -73.443*** | 7.612* |
| | (3.46) | (-2.93) | (1.68) |
| N | 40 | 40 | 40 |

structure, and the regression coefficient is $\beta_{12}$, which is significantly positive at the 1% level. Model (3) shows that the regression coefficient ($\chi_{22}$)of industrial structure upgrading on GTFP is significantly negative at 1% level, but the regression coefficient of digital economy on GTFP ($\chi_{12}$)is significantly positive at 1% level. Further analysis of the mediation effect showed that $\alpha_1$, $\chi_{22}$, $\beta_{12}$ and $\chi_{12}$ were all significant. Therefore, it was believed that other mediation effects might exist. The difference sign between ($\beta_{12}\chi_{22}$) and $\chi_{12}$ indicates that in the whole process of the influence of digital economy on the development of GTFP, the advanced industrial structure plays a certain degree of masking effect, that is, the indirect effect offsets the direct effect of 28.09%. The theory of new structural economics shows that the upgrading of industrial structure plays a positive role when it adapts to the resource endowment at this stage. Otherwise, the "cost disease" of the upgrading of industrial structure will be aggravated if the original balance is destroyed blindly. Therefore, the influence of advanced industrial structure on GTFP is likely to be nonlinear.

At this point, the indirect transmission mechanism of the impact of digital economy on GTFP in textile and apparel industry has been tested. The test results of model (2) in Table 10 and model (2) in Table 11 verify H2. The test results of model (3) in Table 10 and model (3) in Table 11 reject H3. At the same time, digital economy has an impact on GTFP of textile and apparel industry through the rationalization and upgrading of industrial structure. Industrial structure upgrading is the main path for digital economy to promote carbon emission reduction, and H4 has been verified.

## Conclusion and policy recommendations

According to the relevant data of China's textile and apparel industry from 2010 to 2019, this paper measured the GTFP of the textile and apparel industry and its subdivided industries by using the unexpected output super efficiency SBM model and ML index. Then the measurement model was constructed to empirically study the impact of digital economy and industrial structure upgrading on GTFP, and the following conclusions and inspirations were drawn.

1. From 2010 to 2019, the overall GTFP level of the textile and apparel industry is relatively high, and the phenomenon of high energy consumption and high emissions has been significantly improved. Among them, the GTFP of the textile industry has not reached the

optimal production frontier, so management of low-carbon still needs to be strengthened. The GTFP of the textile and apparel industry as a whole is in a stage of low growth, and the growth sources of GTFP of subdivided industries are different. The improvement of GTFP of the textile industry and clothing industry mainly depends on the joint promotion of technical efficiency and technological progress, while the deterioration of GTFP of the chemical fiber manufacturing industry can be improved through the improvement of technical efficiency. Therefore, it is necessary to increase the input of digital technology and comprehensively promote the construction of industrial intelligence, so as to continue to improve the contribution of technological progress and technical efficiency to the GTFP in textile and clothing industries. In addition, it is necessary to pay attention to the optimal allocation of resource elements and the improvement of talent quality, etc. in the chemical fiber manufacturing industry, so as to comprehensively improve its scientific management level, and improve the contribution rate of technical efficiency.

2. Digital economy has a significant role in promoting GTFP, and more efforts should be made to develop digital economy. First, New infrastructure such as data center, Internet and 5G base station should be built under the guidance of the state or through PPP and BOT mode. Through the digital information platform and digital trading platform, the data islands between the textile and apparel industry chain are connected to promote the formation of smart clusters. Second, government should strengthen the R&D investment in digital technology. At the same time, relying on leading enterprises, government can guide small and medium-sized enterprises to actively apply digital information resources, and build a new business form. Third, related industries should accelerate the digital economy' all-round, multi-angle and whole-chain transformation of the textile and apparel industry, strive to transform digital information into advanced productivity, so as to provide an important foundation for the improvement of GTFP.

3. The impact of the digital economy on subdivided industries is significantly different. Specifically, it has a significant promoting effect on the GTFP of the textile industry, but has an insignificant impact on the clothing industry, and has a significant negative effect on the chemical fiber manufacturing industry. Therefore, it is very necessary to gradually implement policies suitable for the development of various industries. The textile industry can further strengthen the degree of integration with digital technology, explore degradable and recyclable fabrics, and take the digital economy as the new engine of industrial development. The government can encourage leading enterprises to play a leading role both online and offline to foster a number of smart parks. In the process of the transformation and development of the clothing industry, the government should formulate and improve relevant preferential policies such as financial support, tax reduction and exemption, technical innovation fund. The clothing industry should also actively develop new technologies, such as the natural degradation technology of clothes, and actively smooth the scene conditions of good linkage with users to publicize environmental awareness. The chemical fiber manufacturing industry relies heavily on petroleum, so it is necessary to control water and gas to force backward production capacity to be eliminated, so that the energy structure can be changed to a green direction. The industry should attach importance to the construction and implementation of the green standardization system, promote and manage "green fiber" trademark, strengthen the binding force of standards, and strictly examine the management of industrial energy conservation and emission reduction. At the same time, an exchange platform for mutual supervision and promotion between enterprises should be established, regularly announcing the energy conservation and emission reduction of enterprises, resolutely prohibiting blind and exemplary behavior.

4. From the perspective of transmission path, digital economy can promote the improvement of GTFP through the rationalization of the industrial structure, and the mediation effect reaches 35.81%. Therefore, it is necessary to gradually change the production and management mode, improve the allocation efficiency of production factors, promote the coordination and linkage of various departments within the industry, and play the dual driving role of industrial structure rationalization and digital economy on GTFP. At the same time, in the process of the influence of digital economy on GTFP, the advanced industrial structure plays a hiding effect of 28.08%. As we know, advanced industrial structure does not necessarily have "structural dividend", and the insufficient or excessive degree of advanced industrial structure may inhibit the growth of GTFP. The conclusions indicate that the penetration of skilled labor and emerging technology is a process of "innovation and destruction". The advancement of the industrial structure is usually accompanied by drastic changes in the industry. If the industrial structure and factor endowments are mismatched, the upgrading of the textile and apparel industry structure is not conducive to the growth of GTFP. Therefore, we must give full consideration to the industrial development situation and the resources endowment, and rationally treat the issue of advanced development.

## Supporting information

**S1 Table. The data of all variables.**
(XLSX)

## Acknowledgments

We would like to thank the reviewers for providing professional comments on the manuscript. We would also like to thank the library of North University of China for providing us with library collection and academic journals.

## Author Contributions

**Conceptualization:** Xiangmei Zhu, Bin Zhang.

**Data curation:** Bin Zhang, Hui Yuan.

**Formal analysis:** Bin Zhang, Hui Yuan.

**Funding acquisition:** Xiangmei Zhu.

**Supervision:** Xiangmei Zhu, Hui Yuan.

**Writing – original draft:** Bin Zhang.

**Writing – review & editing:** Xiangmei Zhu, Bin Zhang.

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
