## [Decision Letter · Decision Letter 0]

17 Aug 2022

PONE-D-22-14541Digital economy, industrial structure upgrading and green total factor productivity——Evidence in textile and apparel industry from ChinaPLOS ONE

Dear Dr. Zhang,

Thank you for submitting your manuscript to PLOS ONE. After careful consideration, we feel that it has merit but does not fully meet PLOS ONE’s publication criteria as it currently stands. Therefore, we invite you to submit a revised version of the manuscript that addresses the points raised during the review process.

We look forward to receiving your revised manuscript.

Kind regards,

Achmad Syafiuddin, Ph.D.

Academic Editor

PLOS ONE

Journal Requirements:

3. We note that you have stated that you will provide repository information for your data at acceptance. Should your manuscript be accepted for publication, we will hold it until you provide the relevant accession numbers or DOIs necessary to access your data. If you wish to make changes to your Data Availability statement, please describe these changes in your cover letter and we will update your Data Availability statement to reflect the information you provide

Reviewers' comments:

Reviewer's Responses to Questions

**Comments to the Author**

1. Is the manuscript technically sound, and do the data support the conclusions?

Reviewer #1: Yes

Reviewer #2: Yes

2. Has the statistical analysis been performed appropriately and rigorously? 

Reviewer #1: Yes

Reviewer #2: Yes

3. Have the authors made all data underlying the findings in their manuscript fully available?

Reviewer #1: Yes

Reviewer #2: Yes

4. Is the manuscript presented in an intelligible fashion and written in standard English?

Reviewer #1: Yes

Reviewer #2: Yes

5. Review Comments to the Author

Reviewer #1: The topic of the research is interesting. Some comments for the paper are:

1). Please write the originality and contribution of the study in the manuscript clearly. I think you can use the following example in this matter ==>

"The originality of this study is... The contribution of this study is..."

2). Is it possible to extend the analysis period (for example, from 2000 through 2020)? Please explain.

3). Please revise the typos in the manuscript.

4). Please avoid to use arrow mark in the part 6 (page 17).

Reviewer #2: We have reviewed this manuscript with the title Digital Economy, Industrial Structure Upgrading and Green Total Factor Productivity- Evidence in textile and apparel industry from china. Please add explanation of the novelty and contribution of the manuscript in abstract. Please give the explanation of the equation number one as well as process to get equation number two, three, and four. Please explain the correlation between table three and equation number five due to there is not symbolic as mention in equation number five.

6. PLOS authors have the option to publish the peer review history of their article (what does this mean?). If published, this will include your full peer review and any attached files.

Reviewer #1: **Yes: **Ubaidillah Zuhdi

Reviewer #2: **Yes: **Mohamad Yusak Anshori

---

## [Author Response · Author response to Decision Letter 0]

15 Sep 2022

Dear Editor:

We appreciate you and reviewers for your precious time and hard work in reviewing our paper and providing professional comments. According to yours and the reviewers’ questions we revised the manuscript and the answers for the comments are listed below:

Response to Editor

Point 1: Please ensure that your manuscript meets PLOS ONE's style requirements, including those for file naming.

Response 1: Thanks for your work in our manuscript. The authors have revised the manuscript format according to the style templates given by PLOS ONE.

Point 2: PLOS requires an ORCID iD for the corresponding author in Editorial Manager on papers 

submitted after December 6th, 2016. Please ensure that you have an ORCID iD and that it is 

validated in Editorial Manager.

Response 2: Thanks for your work in our manuscript. The ORCID ID has been created.

Point 3: We note that you have stated that you will provide repository information for your data at 

acceptance. Should your manuscript be accepted for publication, we will hold it until you provide 

the relevant accession numbers or DOIs necessary to access your data. If you wish to make 

changes to your Data Availability statement, please describe these changes in your cover letter 

and we will update your Data Availability statement to reflect the information you provide

Response 3: Thanks for your work in our manuscript. We make no changes to data availability statement.

Point 4: Please review your reference list to ensure that it is complete and correct. If you have cited 

papers that have been retracted, please include the rationale for doing so in the manuscript text, or remove these references and replace them with relevant current references. Any changes to the reference list should be mentioned in the rebuttal letter that accompanies your revised manuscript. If you need to cite a retracted article, indicate the article’s retracted status in the References list and also include a citation and full reference for the retraction notice.

Response 4: Thanks for your work in our manuscript. The modifications about references are as follows: (1) References with more than six authors list the first six author names, followed by “et al.” (lines 899, 903, 909, 912, 940,976,982). (2) The authors changed the case of letters in the references (lines 962,970,995,1006). In addition, this manuscript does not cite the retracted paper.

Point 5: Please remove your figures/ from within your manuscript file, leaving only the individual TIFF/EPS image files. These will be automatically included in the reviewer’s PDF.

Response 5: Thanks for your question. The authors have removed figures from the manuscript file, and their titles were placed after the paragraph in which figures are first cited. Please review them.

Point 6: Before we can proceed, please clarify which is the accurate Data Availability Statement:

1) "All date files are available from the Statistical Yearbook of China (2010-2019) and Environment Statistical Yearbook of China (2010-2019). They are third party data that anyone can access this data in the same manner as the authors through http://www.stats.gov.cn/tjsj./ndsj/."

2) "All relevant data are within the paper and its Supporting Information files."

Response 6: Thanks for your question. “Data availability statement 1) "means that the original data in the manuscript can be obtained through the website “http://www.stats.gov.cn/tjsj./ndsj/”, but does not include the rest of the calculated data. “Data availability statement 2)” means that all original and processed data in the manuscript can be find from “Supporting Information files". I think “All relevant data are within the paper and its Supporting Information files" is the accurate data availability statement.

Special thanks to you for your good comments.

Notes：Line numbers mentioned in ‘Response to Reviewers’ are based on ‘Revised Manuscript with Track Changes’.

Response to Reviewer 1

Point 1: Please write the originality and contribution of the study in the manuscript clearly. I think 

you can use the following example in this matter ==> "The originality of this study is... The contribution of this study is..."

Response 1: Thanks for your carefully comment. The introduction of originality and contribution needs to be concise and strengthened from the theory and method. Therefore, the authors have written the originality and contribution of the study in the literature review section. (lines 318-337). 

Point 2: Is it possible to extend the analysis period (for example, from 2000 through 2020)? Please explain.

Response 2: Thanks for your valuable comments. Your suggestion is quite important. Extending the research period can indeed improve the accuracy of the results, but it is more difficult to extend the period. The reason is that the data of some measurement indicators of digital economy are missing before 2010. This paper used the digital infrastructure index, digital industrialization index and industrial digitalization index to measure the development of digital economy. The digital infrastructure and digital industrialization data can be traced back to 2000 through the Statistical Yearbook, while the relevant data of industrial digitization index, such as “the number of computers per 100 people, the proportion of enterprises with e-commerce transactions, and e-commerce sales”, have been collected by the National Bureau of Statistics since 2010. The authors tried to use the Wind database and the China Economic Network Statistical Database, and also tried to change the measurement indicators, but none of them can solve this problem, so it is difficult to extend the analysis period.

Point 3: Please revise the typos in the manuscript.

Response 3: Thanks for your carefully reading. The authors have examined the manuscript carefully and the typos have been changed in revised mode (lines 55,56,57,59,62,63,139, 188, 205, 249, 416, 421, 451,462,465,564,664,692,849).

Point 4: Please avoid to use arrow mark in the part 6 (page 17).

Response 4: Thanks for your suggestion. The authors have replaced the arrow mark with literal expression (lines 856, 857, 858, 867,868).

Special thanks to you for your good comments.

Notes：Line numbers mentioned in ‘Response to Reviewers’ are based on ‘Revised Manuscript with Track Changes’.

Response to Reviewer 2

Point 1: Please add explanation of the novelty and contribution of the manuscript in abstract.

Response 1: Thanks for your useful comments. The authors have added the novelty and contribution in abstract. (lines 93-99).

Point 2: Please give the explanation of the equation number one as well as process to get 

equation number two, three, and four.

Response 2: Thanks for your detailed review. You are right, there is indeed a shortage of explanations for the meaning of variables in equation number one, and the authors have completed it (lines427-429, 437-446).

ML index is proposed by Chung in 1997. Equation number two quotes his method, and the authors have made a citation mark in the manuscript in line 453. The ML index can be further decomposed into EC and TC, which are shown in equations number three and four. In order to make the relationship between the ML, EC and TC more clear, the authors made a correction (lines 456,457).

Point 3: Please explain the correlation between table three and equation number five due to there is not symbolic as mention in equation number five.

Response3: Thanks for your good suggestion. Your suggestion is quite important. The comprehensive index of digital economy development (column two in table three) and the sub-index of digital economy development (column three, four and five in table three, corresponding to digital infrastructure index, digital industrialization index and industrial digitalization index) can be obtained according to equations number five. The authors have explained the relevant symbols appearing in table three (lines 487-489). 

Special thanks to you for your good comments.

Notes：Line numbers mentioned in ‘Response to Reviewers’ are based on ‘Revised Manuscript with Track Changes’.

Thanks again for yours and the reviewers’ valuable comments and hard work. 

Kind regards,

Bin Zhang

1158597163@qq.com

Department of Economics and Management

North University of China

---

## [Editor Report · Decision Letter 1]

24 Oct 2022

Digital economy, industrial structure upgrading and green total factor productivity——Evidence in textile and apparel industry from China

PONE-D-22-14541R1

Dear Dr. Zhang,

We’re pleased to inform you that your manuscript has been judged scientifically suitable for publication and will be formally accepted for publication once it meets all outstanding technical requirements.

Kind regards,

Achmad Syafiuddin, Ph.D.

Academic Editor

PLOS ONE
---

## [Editor Report · Acceptance letter]

27 Oct 2022

PONE-D-22-14541R1 

Digital economy, industrial structure upgrading and green total factor productivity——Evidence in textile and apparel industry from China 

Dear Dr. Zhang:

I'm pleased to inform you that your manuscript has been deemed suitable for publication in PLOS ONE. Congratulations! Your manuscript is now with our production department. 

Kind regards, 

on behalf of

Dr. Achmad Syafiuddin 

Academic Editor

PLOS ONE